# The Weight-Based Feature Selection (WBFS) Algorithm Classifies Lung Cancer Subtypes Using Proteomic Data

**DOI:** 10.3390/e25071003

**Published:** 2023-06-29

**Authors:** Yangyang Wang, Xiaoguang Gao, Xinxin Ru, Pengzhan Sun, Jihan Wang

**Affiliations:** 1School of Electronics and Information, Northwestern Polytechnical University, Xi’an 710129, China; wangyang2154@mail.nwpu.edu.cn (Y.W.);; 2Xi’an Key Laboratory of Stem Cell and Regenerative Medicine, Institute of Medical Research, Northwestern Polytechnical University, Xi’an 710072, China

**Keywords:** feature selection, information theory, The Cancer Proteome Atlas (TCPA), The Cancer Genome Atlas (TCGA), lung cancer, Bayesian network, biomarkers

## Abstract

Feature selection plays an important role in improving the performance of classification or reducing the dimensionality of high-dimensional datasets, such as high-throughput genomics/proteomics data in bioinformatics. As a popular approach with computational efficiency and scalability, information theory has been widely incorporated into feature selection. In this study, we propose a unique weight-based feature selection (WBFS) algorithm that assesses selected features and candidate features to identify the key protein biomarkers for classifying lung cancer subtypes from The Cancer Proteome Atlas (TCPA) database and we further explored the survival analysis between selected biomarkers and subtypes of lung cancer. Results show good performance of the combination of our WBFS method and Bayesian network for mining potential biomarkers. These candidate signatures have valuable biological significance in tumor classification and patient survival analysis. Taken together, this study proposes the WBFS method that helps to explore candidate biomarkers from biomedical datasets and provides useful information for tumor diagnosis or therapy strategies.

## 1. Introduction

Lung cancer is one of the deadliest cancers in the world, and lung adenocarcinoma (LUAD) and lung squamous cell carcinoma (LUSC) are the most common subtypes, which have drastically different biological signatures [1]. The precise biomarkers to be implemented in clinical settings remain ambiguous, and accurate classification of lung cancers into clinically significant subtypes is of utmost importance in making therapeutic decisions; this gap demands immediate attention.

The Cancer Proteome Atlas (TCPA) database [2] is a large-scale proteomic database that contains molecular and clinical data from over 11,000 patient samples across 32 different cancer types. The Cancer Genome Atlas (TCGA) database [3,4] is a comprehensive genomic database that also contains the clinical information of patients. Both TCPA and TCGA databases are publicly available and have been widely used by researchers around the world. They provide valuable resources for cancer research, enabling researchers to investigate the molecular mechanisms underlying cancer and identify new targets for therapeutic intervention. However, discovering key biomarkers for lung cancer from the extensive datasets in these databases is a difficult task in biomedical research [5].

Feature selection is an important task in machine learning and data analysis, where the goal is to identify a subset of relevant features from a large set of potential predictors [6]. One approach to feature selection is based on information theory; such an approach involves quantifying the amount of information that a feature provides about the outcome variable of interest. This approach is especially useful in situations where the number of potential predictors is large and the relationship between the predictors and the outcome variable is complex. Information-theoretic feature selection methods are based on measures such as entropy, mutual information, and conditional mutual information. These measures are used to assess the relevance of a feature to the outcome variable, as well as the redundancy between features. The goal is to select a subset of features that maximizes the amount of relevant information while minimizing the amount of redundant information.

One application of information-theoretic feature selection is in the field of biomarker discovery, where the goal is to identify a set of molecular features (such as genes or proteins) that are associated with a particular disease or condition [7]. By using information-theoretic methods to select key biomarkers, researchers can gain insights into the underlying biological mechanisms of the disease, as well as develop diagnostic and therapeutic tools. In the past two decades, many studies on filter feature selection algorithms based on information theory have been reported due to their robustness and computational efficiency [8]. The pioneering work of Battiti employed the mutual information (MI) criterion to evaluate a set of candidate features, and his research proved that mutual information is suitable for assessing arbitrary dependencies between random variables [8,9]. To date, most feature selection methods based on information theory obtain the optimal feature subset by maximizing correlation and minimizing redundancy, such as Mutual Information Maximisation (MIM) [10], Mutual Information Based Feature Selection (MIFS) [9], Mutual Information Feature Selector under Uniform Information Distribution (MIFS-U) [11], Max-Relevance and Min-Redundancy (mRMR) [12], Conditional Informative Feature Extraction (CIFE) [13], and Conditional Redundancy (CONDRED) [14]. Most existing feature selection strategies aim to maximize the classification performance only by considering the relevance between features and classes and redundancy between pairs of features without assessing interactions and complementarity; consequently, some features with discriminative ability may be mistakenly removed. The concept of interaction has been widely explored in different scenarios [15]. For example, exploring the interaction information of microarray gene expression data is beneficial for discovering cancer biomarkers and identifying cancer subtype classifications [16].

This study involved an extensive examination of a lung cancer protein expression dataset sourced from the TCPA database, in conjunction with phenotype and survival information from the TCGA database. Firstly, it aimed to provide an overview of the functional proteomic heterogeneity of LUAD and LUSC tumor samples. Secondly, a unique filter weight-based feature selection (WBFS) method that assesses selected features and candidate features was used to identify candidate protein biomarkers that exhibited superior performance in classifying the two major lung cancer subtypes. Additionally, Bayesian etworks (BNs) were utilized to identify the direct impact factors that had causal relationships with the classification of the two subtypes. Finally, the potential clinical implications of the candidate protein biomarkers for prognosis were evaluated.

## 2. Materials and Methods

### 2.1. Data Acquisition and Preprocessing

We extracted the lung cancer subset from the original dataset (TCGA-PANCAN32-L4.zip belongs to the level 4 data set, and the batch effect has been processed in the past) in TCPA (https://tcpaportal.org/tcpa, accessed on 11 January 2023), resulting in 687 tumor cases comprising 362 LUAD samples and 325 LUSC samples, with 258 proteins. The proteins that contained missing values (“NA”) in over 50% of the samples were removed, and the “NA” values for six proteins were replaced by their average values. This led to the creation of a proteome profiling dataset consisting of 687 tumor samples and 217 proteins. We also downloaded relevant clinical information, including survival information and phenotype information, for these 687 tumor samples from the TCGA data (11 January 2023).

### 2.2. Proteome Profiling Analysis

T-Distributed Stochastic Neighbor Embedding (t-SNE) is a non-linear technique that is particularly useful for visualizing high-dimensional data in low-dimensional space [17]. Principal Component Analysis (PCA) is a linear method that aims to reduce the number of features in a dataset while retaining the maximum amount of information. It can be used to capture the most important patterns in the data and visualization of the classification performance. Unlike PCA, t-SNE is not a linear method and instead tries to preserve the local structure of the data in the low-dimensional space.

We utilized t-SNE and PCA techniques based on the “Rtsne” [18] and “FactoMineR” [19] packages to visualize the differences between the original proteins and the selected proteomic biomarkers.

### 2.3. Using WBFS to Obtain Candidate Protein Biomarkers

High-dimensional datasets have stimulated the development of feature selection techniques. In this section, some basics of information theory are outlined, and then a new feature selection algorithm based on information theory is proposed.

Mutual information is the overlapping information of two entropies and is commonly utilized to analyze the correlation and statistical dependency between two random variables. It has been employed in various feature selection algorithms for obtaining the redundancy between the input features X and the classifications Y as well as the correlation between any pair of features.
(1)$$I(X;Y)=\sum_{x\in X}\sum_{y\in Y}p(x,y)\log\frac{p(x,y)}{p(x)p(y)}$$
where p(x) is the probability when X=x, and p(x,y) is the joint probability when X=x,Y=y.

Conditional mutual information is the mutual information of X and Y given the third variable Z and can be computed by:(2)$$I(X;Y|Z)=\sum_{z\in Z}p(z_{k})\sum_{x\in X}\sum_{y\in Y}p(x,y|z)\log\frac{p(x,y|z)}{p(x|z)p(y|z)}$$
where p(x|z) is the conditional probability of X=x when Z=z is known, and p(x,y|z) is the conditional probability of X=x,Y=y when Z=z is known.

As an extension of mutual information, 3-way interaction information denotes the information shared by three random variables X,Y,Z and can be determined using conditional information and mutual information:(3)I(X;Y;Z)=I(X,Y;Z)−I(X;Z)−I(Y;Z)

Interaction information can be positive, zero, or negative; it is positive when the two variables of X,Y together provide information that cannot be provided by any one of them individually. Therefore, I(X;Z)<I(X;Z|Y) is always true when X,Y have an interaction relationship.

Table 1 summarizes all symbols used for the WBFS formula derivation.

For a dataset D that has m features $F=\{f_{1},f_{2},\ldots ,f_{m}\}$ and n samples, the aim of a feature selection method is to find an optimal feature subset $F^{\prime }=\{f_{1},f_{2},\ldots ,f_{k}\}$ that can maximize the objective function $J(f_{k})$ and yield the same or a better classification accuracy than the original feature set as much as possible, where k≤m and $F^{\prime }\subseteq F$. The concepts of relevance, redundancy, complementarity, and interaction are critical for a feature selection approach. The relevance between feature and class can be expressed using the mutual information of $I(f_{i};C)$; $f_{i}$ is called a stronger correlated feature than $f_{j}$ when $I(f_{i};C)>I(f_{j};C)$. In addition, the mutual information $I(f_{i};f_{j})$ between pairs of features is called redundancy. Considering correlation alone is definitely not sufficient because redundant features may reduce the classification performance and increase computational complexity; therefore, the redundancy should be eliminated while a new candidate feature $f_{k}$ is selected into S from the candidate feature set F/S. However, there is still a flaw in the objective function $J(f_{k})$ based only on relevance and redundancy due to the absence of supplementary information when a new candidate feature $f_{k}$ is introduced into S.

We used the following equation as the objective function to screen the candidate features individually:(4)$$J(f_{k})=\arg\max\sum_{f_{k}\in F-S,f_{j}\in S}\omega *I(f_{k};C|f_{j})$$

Specifically, instead of using equal weight to define the influence of selected features on candidate features, a dynamic parameter ω is considered to evaluate the differences of the selected features and updated when a new feature $f_{k}$ is introduced into S.

For two interaction features, $f_{k}$ and $f_{j}$, $I(f_{j};C)<I(f_{j};C|f_{k})$ will be true if their interaction information is positive. In other words, the addition of feature $f_{k}$ will produce a positive influence in predicting C for $f_{j}$. Hence, we can use $I(f_{k};C|f_{j})-I(f_{k};C)$ to measure the weight ω, which can be expressed as follows:(5)$$\omega =1+\frac{2(I(f_{j};C|f_{k})-I(f_{j};C))}{H(f_{j})+H(C)}$$

**Theorem** **1.**0≤ω<2.

**Proof of Theorem** **1.**For feature $f_{k}$, $f_{j}$ and class C, we have:$$\begin{array}{l} \left\{\begin{array}{c} 0\leq I(f_{j};C)\leq H(f_{j})\\ 0\leq I(f_{j};C)\leq H(C) \end{array}\right.\Rightarrow 0\leq 2I(f_{j};C)\leq H(f_{j})+H(C)\\ \left\{\begin{array}{c} 0\leq I(f_{j};C|f_{k})\leq H(f_{j})\\ 0\leq I(f_{j};C|f_{k})\leq H(C) \end{array}\right.\Rightarrow 0\leq 2I(f_{j};C|f_{k})\leq H(f_{j})+H(C) \end{array}$$
Then,
$$\left\{\begin{array}{c} 0\leq \frac{2I(f_{j};C)}{H(f_{j})+H(C)}\leq 1\\ 0\leq \frac{2I(f_{j};C|f_{k})}{H(f_{j})+H(C)}\leq 1 \end{array}\right.\Rightarrow -1\leq \frac{2(I(f_{j};C|f_{k})-I(f_{j};C))}{H(f_{j})+H(C)}\leq 1$$
Hence,
$$0\leq \omega =1+\frac{2(I(f_{j};C|f_{k})-I(f_{j};C))}{H(f_{j})+H(C)}\leq 2$$ □

Based on Equations (4) and (5), the final objective function of WBFS is shown as:(6)$$J_{WBFS}(f_{k})=\arg\max\sum_{f_{k}\in F-S,f_{j}\in S}\left(1+\frac{2(I(f_{j};C|f_{k})-I(f_{j};C))}{H(f_{j})+H(C)})*I(f_{k};C|f_{j}\right)$$

Our proposed method of WBFS consists of two components: ω and $I(f_{k};C|f_{j})$. The first component considers not only the importance of features in S but also the influence of candidate feature $f_{k}\in F/S$ on the relevance between S and C. The strong positive interaction of features $f_{k}\in F/S$ and $f_{j}\in S$, the larger this value. The second component, $I(f_{k};C|f_{j})$, is conditional mutual information, which fully covers the relevance between candidate feature $f_{k}$ and class C based on the selected feature $f_{j}\in S$. Algorithm 1 presents the pseudocode of the WBFS algorithm:
**Algorithm 1** Weight-based feature selection (WBFS)$Input:\;F,\;S,\;C,\;K\;\left(K\;\leq \;\left|F\right|\right)$$Output:\;F^{\prime }$$Initialization:\;S\leftarrow \;\emptyset ,\;F^{\prime }\leftarrow \;\emptyset ;$$for\;i=0;i<\left|F\right|;i++do$  
$MI_{i}\leftarrow \;I\left(f_{i};C\right);$
end for$S\leftarrow f^{\prime }\;\;which\;I\left(f^{\prime };C\right)=\max\left(MI\right);$$\left|S\right|\leftarrow 1;$$while\;\left|S\right|<K\;\;do$  
$for\;k=0;k<\left|F\backslash S\right|;k++do$
    
$J_{WBFS}\left(f_{k}\right)=argmax\underset{f_{k}\in F-S,f_{j}\in S}{\sum }\left(1+\frac{2*\left(I(f_{j};C|f_{k})-I\left(f_{j};C\right)\right)}{H\left(f_{j}\right)+H\left(C\right)}\right)*I(f_{k};C|f_{j});$
    $S\leftarrow f_{k}$;  
end for
end while$F^{\prime }\leftarrow S;$$Return\;F^{\prime };$

In Algorithm 1, $H(f_{j})$ and H(C) represent the entropy of the feature $f_{j}$ and the categorical variable C, respectively.

When we need to select the top K features from the original feature set F, the steps of the WBFS algorithm are as follows:

First, we obtain the mutual information between each feature in F and class C. Second, the feature $f^{\prime }$, which has the largest relevance among the features, is added into S. Third, for the candidate feature set F/S, Equation (6) is utilized to obtain the feature $f_{k}$ and add it to the selected feature set S. When the number of features in S is equal to K, S is the optimal feature set $F^{\prime }$.

The time consumption largely depends on the computation times of mutual information or conditional mutual information for information theory-based feature selection methods. For a dataset D with m samples and n features, the time complexity is O(m) when mutual information or conditional mutual information is calculated. When a new feature is introduced into the selected feature set, mutual information or conditional mutual information among all features must be calculated; the time complexity is O(mn) for one iteration. Therefore, for WBFS, the final time complexity is O(kmn) if the top k features are chosen from the original features. The MIFS [9], mRMR [12], CIFE [13] and CONDRED [14] methods have the same time complexity as WBFS; and the time complexity of MIM [10] is O(mn) because only the relevance between features and classes is considered.

### 2.4. Using Bayesian Networks to Discover Causalities

Causal discovery aims to find causal relations by analyzing observational data [20]. As a widely used graphical model in bioinformatics and employed for related aspects of classification and prediction, the Bayesian network (BN) provides a convenient and coherent way to represent uncertainty from a perspective of probability [21]. BN models can estimate the joint probability distribution P over a vector of random variables $X=\left(X_{1},\ldots ,X_{n}\right)$, and the joint probability distribution factorized as a product of several conditional distributions denotes the dependency or independency structure by a directed acyclic graph (DAG):(7)$$P(X_{1},\ldots ,X_{n})=\prod_{i=1}^{n}P(X_{i}|Pa(X_{i}^{G}))$$
where $P(X_{1},\ldots ,X_{n})$ is the joint probability, $Pa(X_{i}^{G})$ are the parent nodes of $X_{i}$ and $P(X_{i}|Pa(X_{i}^{G}))$ is the conditional probability of $X_{i}$ when the parent nodes of $X_{i}$ are known. Tsamardinos et al. were the first to establish the connection between local causal discovery and feature selection [22]. Yu et al. developed a unified view of causal and non-causal feature selection methods based on the BN perspective, and their experimental results show that non-causal feature selection is more suitable for high-dimensional data, especially for gene sequencing data [23]. Therefore, the combination of non-causal feature selection and BN local causal discovery is a promising strategy for identifying key signatures of a target in a high-dimensional dataset. The regulatory network composed of the selected biomarkers defines the regulatory interactions among regulators and their potential targets and can be used for inference and prediction. We constructed a regulatory network using BN to explore the causality interaction between any two of the selected proteins using WBFS based on the Causal Learner toolbox, which is available on GitHub (https://github.com/z-dragonl/Causal-Learner, accessed on 11 January 2023) [24].

### 2.5. Receiver Operating Characteristic (ROC) and Survival Analysis

The candidate protein biomarkers obtained with WBFS were analyzed for ROC and survival curves in order to evaluate the potential applications of these candidate signatures as possible markers for lung cancer diagnosis and prognosis. Medcalc statistical software [25] was used to analyze the candidate proteins for distinguishing LUAD and LUSC samples, and the sensitivity, specificity, and area under the curve (AUC) values were obtained. A multiple regression model to evaluate the combination classification performance of specific set biomarkers was also assessed, and it can be expressed as follows:(8)$$multiple\;\mathrm{regression}=\frac{1}{1+e^{-(cons\tan t+\sum_{i=1}^{n}coefficient_{i}*\exp ression_{i})}}$$

In the above Equation (8), the constant represents the value that would be predicted for the dependent variable if all the independent variables were simultaneously equal to zero, the $coefficient_{i}$ is the slope of the linear relationship of the i-th protein, and the $expression_{i}$ is the expression value of the i-th protein.

For survival analysis, Kaplan–Meier survival curves were explored and visualized (using α = 0.05 as the test level) using the Survminer toolkit [26] in R based on age, gender, and the tumor, node, metastasis (TNM) stage (clinical information was collected in the TCGA dataset), as well as the expression pattern of candidate proteins in LUAD and LUSC subjects.

## 3. Results

### 3.1. The Distributions Overview of the LUAD and LUSC Tumor Samples

The t-stochastic neighbor embedding (t-SNE) algorithm and principal component analysis (PCA), using the “Rtsne” and “FactoMineR” packages, respectively, were employed to display the difference between the original proteins and the 10 proteomic biomarkers. Overviews of the original lung cancer dataset based on t-SNE and PCA are shown in Figure 1A,C, respectively. t-SNE and PCA were also performed based on the subset containing 10 signatures. As shown in Figure 1B, the samples using 10 protein signatures clustered significantly better than the 258 original proteins. From Figure 1C,D, we also observed that the PCA scores for Dim1 significantly increased from 14.4% to 36.4%.

### 3.2. Using WBFS Method Identifying Protein Signatures for Classifying the Two Cancer Subtypes

#### 3.2.1. Evaluate WBFS Classification Performance Based on UCI Datasets

To evaluate the classification performance of the WBFS algorithm and the advantage in terms of prediction accuracy compared with other feature selection algorithms, 12 high-dimensional datasets with different numbers of features, samples, and classification labels from the UCI repository (https://archive.ics.uci.edu/mL/datasets.php, accessed on 11 January 2023) and additional studies were employed (https://github.com/jundongl/scikit-feature, accessed on 11 January 2023). We discretize continuous datasets using an equal-width strategy into five bins. Additionally, we introduced the ratio concept to evaluate the classification difficulty for each dataset, with a smaller value indicating a more challenging feature selection. For dataset D, N is the number of samples, m is the median arity of the features, and c is the number of classes. The ratio can be calculated by N/(mc). All 12 datasets are described in Table 2.

The experiment was performed using MATLAB 2020 based on Windows 10 with a 1.6 GHz Intel Core CPU and 8 GB memory. Three types of classifiers, k-nearest neighbors (KNN) [27], naïve Bayes classifier (NBC) [28], and the Library for Support Vector Machines (LibSVM) [29], were used to test the prediction performance using selected features in the experiment. Ten-fold cross-validation was employed, in which each of these datasets was randomly partitioned into a training dataset (90%) and a testing dataset (10%) and fed to every feature selection algorithm 10 times. Five well-known feature selection algorithms were selected to be compared with the proposed WBFS method, including CONDRED [14], MIM [10], mRMR [12], DISR [30], and CIFE [13]. A *t*-test was used to determine the significant differences between two groups of classification accuracies, and a Win/Tie/Loss (W/T/L) paradigm was adopted for describing performance: Win means WBFS shows better than other baseline methods, Tie means that there is no statistically significant difference with other methods and other cases are classified as Loss. Specifically, the ‘*’ symbols and ‘v’ identify statistically significant (at the 0.1 level) wins or losses over the proposed WBFS method. The average accuracies, standard deviation, and W/T/L information of 12 datasets based on three classifiers are shown in Table 3, Table 4 and Table 5. The number of selected features for all methods was fixed at 15 and k=5 for the KNN classifier.

In Table 3, Table 4 and Table 5 bold values indicate the largest value among the six feature selection methods. WBFS obtains the best average classification accuracies of 84.48%, 73.15%, and 83.01% and occupies first place on five datasets (Lung, Krvskp, Madelon, Musk, and Waveform), six datasets (Breast, Colon, Ionosphere, Krvskp, Sonar, Waveform), and six datasets (Breast, Krvskp, Madelon, Musk, Sonar, Waveform) based on the three classifiers KNN, NB and LibSVM, respectively. For the other methods of CONDRED [14], MIM [10], mRMR [12], DISR [30], and CIFE [13], the numbers of cases for which they achieve the highest classification accuracy are 1, 2, 2, 1, 1 with KNN; 1, 1, 0, 1, 3 with NBC; and 1, 1, 2, 1, 1 with LibSVM, respectively.

From the perspective of W/T/L, WBFS obtains the best result in most cases. According to Table 5, WBFS achieves significantly higher classification accuracy than the CONDRED, MIM, mRMR, DISR, and CIFE methods with case numbers of 7, 6, 3, 6, and 8. For Table 3 and Table 4, the numbers are 6, 5, 3, 5, 7 and 7, 4, 3, 4, 4.

Considering that the above six feature selection algorithms are all based on a feature ranking strategy, an effective method for further comparing their classification accuracies is to individually add features for learning. We employed this strategy using a range of 1–30 selected features. Figure 2 shows the comparative results based on six representative datasets (Lung, Madelon, Sonar, Musk, Krvskp and Waveform). For each subgraph, the x-axis refers to the first k selected features, and the y-axis represents the average classification accuracy of three classifiers for the top k selected features. In Figure 2, it is obvious that the proposed WBFS method shows better average accuracies than the others in most cases based on the Lung, Madelon, Sonar, and Musk datasets. For instance, WBFS obtains the highest average classification accuracy of 76.39% with only seven features, while accuracies of 68.50%, 67.44%, 70.40%, 74.31%, and 64.84% are achieved by MIM, mRMR, DISR, CIFE and CONDRED, respectively, with the same number of selected features. For Krvskp and Waveform, the accuracies of each method are comparable because their ratio value is sufficiently large; WBFS still obtains the best accuracy in most cases.

In addition, we also compared the classification accuracy of feature selection subsets obtained by the WBFS algorithm and ensemble learning algorithm Random Forest (RF) based on the Isolet, Sonar, Waveform, Madelon, and Splice datasets, as shown in Appendix A. The results suggest that both methods have similar classification accuracy on low-dimensional datasets such as Waveform, Sonar, and Splice. However, WBFS demonstrates a clear advantage in classification accuracy on high-dimensional datasets such as Madelon and Isolet.

All of the above experiments demonstrate that the WBFS technique performs better than the other methods in terms of classification accuracy, which lays the foundation for its application in bioinformatics data.

#### 3.2.2. Using WBFS to Obtain the Top 10 Candidate Biomarkers

We discretized the expression data into three bins by utilizing an equal-width strategy for WBFS according to the three expression states of protein (high expression, normal expression, and low expression). The top 10 proteins (MIG6, GAPDH, NDRG1_pT346, BRD4, CD26, TFRC, INPP4B, GSK3ALPHABETA, IGFBP2, and DUSP4) were screened by applying the WBFS algorithm, and they can be regarded as the candidate biomarkers that are most closely associated with the classification of lung cancer subtypes.

### 3.3. Using Bayesian Networks to Discover Causalities

In Figure 3, we can observe that the direct determinants for differentiating the subtypes of lung cancer include the protein markers MIG6, NDRG1_pT346, BRD4, CD26, INPP4B, and DUSP4, while the others can be considered as having an indirect impact on the classification outcome. Therefore, the Bayesian network (BN) provides significant factors for mechanistic studies at the molecular level.

### 3.4. ROC and Survival Analysis of the Candidate Protein Signatures

From Figure 4 and Table 6, it is possible to accurately distinguish LUAD and LUSC by each of the 10 protein biomarkers, as indicated by the AUC, which had a value up to 0.72 (*p* < 0.001). Multiple regression results of the 10 biomarkers indicated that eight protein markers, MIG6, GAPDH, NDRG1_pT346, BRD4, CD26, TFRC, INPP4B, and DUSP4, had significant differentiation abilities in disease classification between the LUSC and LUAD groups, as shown in Appendix A. The detailed results of multiple regression analysis are shown in Table 6. The combination of the eight features achieved the best classification performance (AUC = 0.960, *p* < 0.001).

Figure 5 shows the Kaplan–Meier survival curves of the above factors, which demonstrates that MIG6, TFRC, INPP4B and IGFBP2 factors have a significant impact on the overall survival rate of LUAD patients (*p* < 0.05), and the MIG6, TFRC, and INPP4B factors have a significant negative prognosis correlation (*p* < 0.05). For LUSC patients, the factors of MIG6, GAPDH, CD26, and TFRC are all significantly correlated with the overall survival rate of patients (*p* < 0.05), and TFRC was the only factor that has a significant positive prognosis correlation (*p* < 0.05).

## 4. Discussion

Lung cancer is one of the most common malignant tumors worldwide. Accurate classification of lung cancers into clinically significant subtypes is of utmost importance, making it critical for clinical and scientific researchers to discover new molecular markers and potential diagnostic and therapeutic targets [6,31]. Additionally, it is worth noting that the TCPA platform has already standardized protein expression information, while the TCGA database catalogs and explores cancer-causing genomic alterations, establishing a comprehensive “atlas” of cancer genomic signatures. By integrating the information from TCPA and TCGA, we could identify potential diagnostic and prognostic biomarkers for lung tumor subtypes to help understand the underlying mechanisms of tumorigenesis and improve approaches or standards for cancer diagnosis and therapy.

Feature selection is a fundamental technique for achieving efficient data reduction in high-dimensional datasets, such as those used in biological and text data mining [8,32,33]. This strategy is crucial for identifying accurate models and key factors in classification or clustering tasks [34,35]. Many studies have focused on developing feature selection strategies that return an optimal feature subset. Information theory-based approaches, such as mutual information and its extensions, can efficiently evaluate the relevance, redundancy, or interaction information [36,37,38]. However, most proposed approaches concentrate primarily on relevance or redundancy without considering interaction information. In this paper, we present a new weight-based feature selection (WBFS) method that considers the weight between selected features and candidate features based on information theory. Our proposed algorithm outperforms other methods in terms of classification performance, as demonstrated through experiments on 12 datasets and three classifiers.

Feature selection methods have an interesting application in biomarker discovery from high-throughput genomics data, which can guide clinical diagnosis by identifying the most discriminative biomarkers for a particular problem. In this study, the WBFS method was applied to real-world lung cancer datasets to obtain key protein biomarkers for the classification of lung tumor subtypes. The causalities among the selected biomarkers were also explored using the Bayesian network (BN) strategy. The study found six promising protein biomarkers that were directly correlated with the classification of lung cancer subtypes, including MIG6, NDRG1_pT346, BRD4, CD26, INPP4B, and DUSP4. ROC analysis and survival curves based on selected proteins were also implemented. The ROC analysis has shown that 10 protein biomarkers were able to accurately distinguish between LUAD and LUSC with an AUC value of up to 0.72 and that eight of the protein markers had significant utility in disease classification. When combined, the eight features produced an AUC of 0.960, indicating strong classification performance. Survival curves showed that MIG6, TFRC, INPP4B, and IGFBP2 had a significant impact on the overall survival rate of LUAD patients, while MIG6, GAPDH, CD26, and TFRC were all significantly correlated with the overall survival rate of LUSC patients. A study [39] demonstrated that MIG6 is essential for controlling the regulation of cell signaling in lung cancer cells.

This work involves some limitations that are worth noting. Firstly, our WBFS method is based on the feature ranking strategy, where the selection of the number of features k is critical for achieving optimal classification performance and accuracy. Unfortunately, there is currently no effective strategy for determining the optimal value of k. Moreover, in the era of big data, it is more important than ever to identify the causal relationships among large datasets. However, most of the proposed feature selection methods are non-causal, and thus, it is necessary to establish a connection between non-causal feature selection and causality inference. In our future work, we plan to focus on two main aspects:

(1)Developing a more efficient feature selection algorithm that can automatically determine the optimal number of selected features based on the intrinsic dimension of a high-dimensional dataset.(2)Investigating the relationship between causal and non-causal feature selection methods and applying non-causal feature selection methods to the Bayesian network. This will facilitate Bayesian network structure learning based on high-dimensional data.

## 5. Conclusions

In this paper, we developed a new weight-based feature selection (WBFS) method to identify protein biomarkers to distinguish between two different subtypes (LUAD and LUSC) of lung cancer. Additionally, we further explored the direct influencing factors for lung cancer subtype classification using Bayesian networks. Six promising protein biomarkers (MIG6, NDRG1_pT346, BRD4, CD26, INPP4B, and DUSP4) specifically contributed to the subtype classification. These biomarkers, along with others, were able to accurately distinguish between LUAD and LUSC with a high AUC value. The identification and characterization of these biomarkers hold enormous potential for improving the diagnosis, treatment, and outcomes of lung cancer patients.

## Figures and Tables

**Figure 1 entropy-25-01003-f001:**
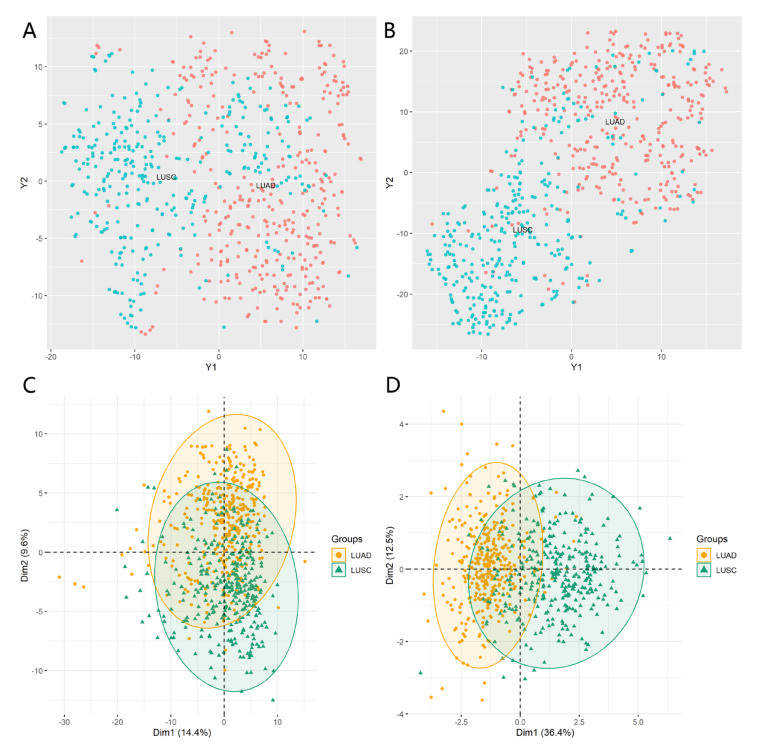
Overview of the proteome profiling across lung adenocarcinoma (LUAD) and lung squamous cell carcinoma (LUSC) samples. The subgraphs (**A**,**B**) are the results performing t-SNE; the subgraphs (**C**,**D**) are the results performing PCA; (**A**) t-SNE and (**C**) PCA showed the clusters of samples based on whole proteome profiling; (**B**) t-SNE and (**D**) PCA showed the clusters of samples based on the top 10 selected protein profiling.

**Figure 2 entropy-25-01003-f002:**
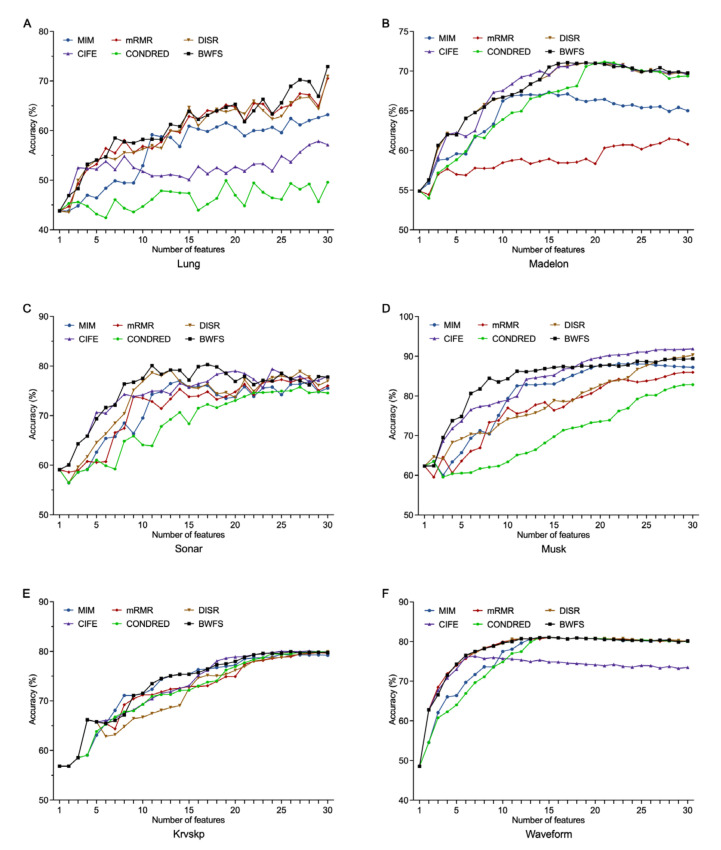
Average classification accuracy based on three classifiers vs. different number of selected features for six datasets: (**A**) Lung, (**B**) Madelon, (**C**) Sonar, (**D**) Musk, (**E**) Krvskp, (**F**) Waveform.

**Figure 3 entropy-25-01003-f003:**
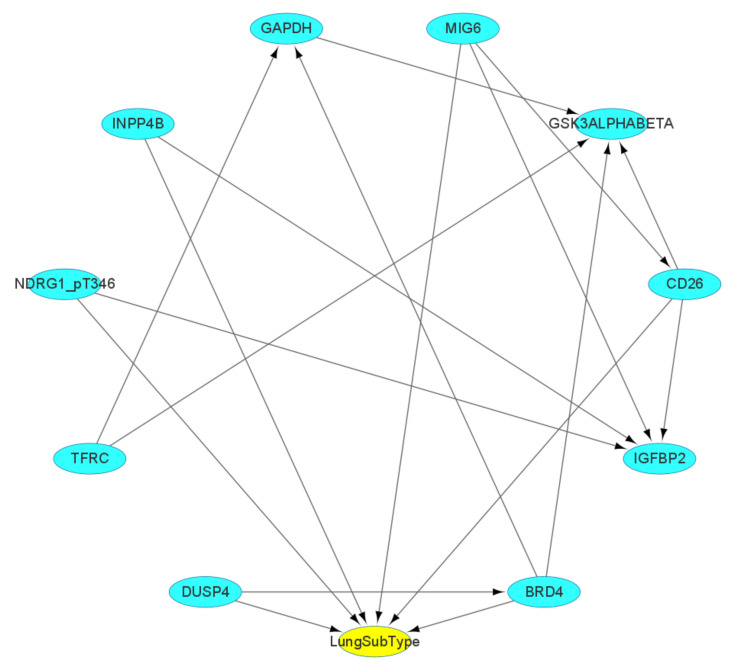
The Bayesian network (BN) for the top 10 selected signatures and class label based on lung tumor dataset. These directed arrows represent the causal relationship between every two nodes. The cyan nodes are the key factors that are critical for subtype classification, and the yellow node represents the lung subtype classification.

**Figure 4 entropy-25-01003-f004:**
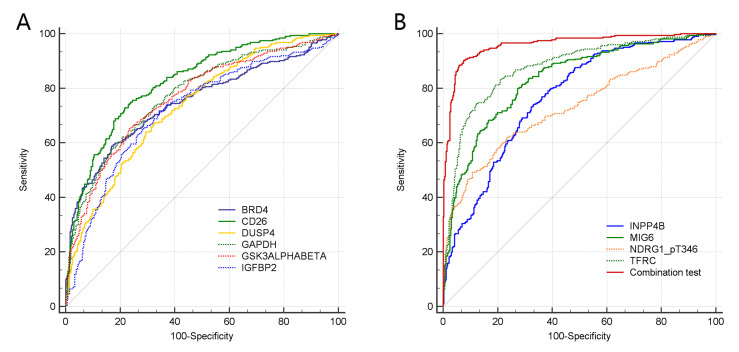
ROC curves of the 10 selected protein features and the combination test of 8 biomarkers. (**A**) shows the ROC curves of the 6 protein features of BRD4, CD26, DUSP4, GAPDH, GSK3ALPHABETA and IGFBP2; (**B**) shows the ROC curves of the 4 protein features of INPP4B, MIG6, NDRG1_pT346, TFRC and the combination test curve of 8 biomarkers.

**Figure 5 entropy-25-01003-f005:**
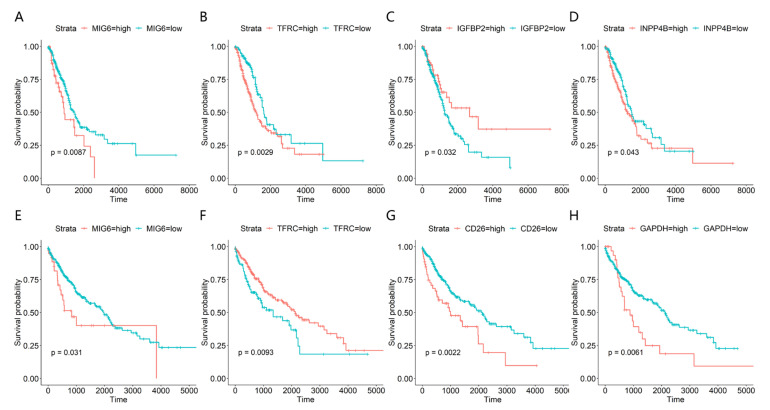
The Kaplan–Meier curves for lung adenocarcinoma (LUAD) (**A**–**D**) and lung squamous cell carcinoma (LUSC) (**E**–**H**). The values of all protein expressions were divided into high and low groups by cut-off values. The horizontal axis represents the survival time (day), and the vertical axis represents the overall survival rate.

**Table 1 entropy-25-01003-t001:** The symbols used in feature selection method.

Symbols	Description	Symbols	Description
D	Dataset	n	The sample size of D
F	Original feature set	m	The feature size of F
C	Class labels	|S|	The feature size of S
S	The selected feature subset	K	The size of $F^{\prime }$
$F^{\prime }$	Optimal feature subset	$f_{j}$	Feature number j
$f_{i}$	Feature number i	F/S	The candidate feature set
$f_{k}$	The candidate feature	$J(f_{k})$	The objective function

**Table 2 entropy-25-01003-t002:** Description of 12 datasets for experiment.

Dataset Name.	Dataset Source	# Instances	# Features	# Classes	Ratio
Lung	https://github.com/jundongl/scikit-feature (accessed on 11 January 2023)	73	325	7	4
Colon	https://github.com/jundongl/scikit-feature (accessed on 11 January 2023)	62	2000	2	10
Isolet	https://github.com/jundongl/scikit-feature (accessed on 11 January 2023)	1560	617	26	12
Sonar	https://archive.ics.uci.edu/dataset/151/connectionist+bench+sonar+mines+vs+rocks (accessed on 11 January 2023)	208	60	2	20
Ionosphere	https://archive.ics.uci.edu/dataset/52/ionosphere (accessed on 11 January 2023)	351	34	2	35
Breast	https://archive.ics.uci.edu/dataset/17/breast+cancer+wisconsin+diagnostic (accessed on 11 January 2023)	569	30	2	57
Landsat	https://archive.ics.uci.edu/dataset/146/statlog+landsat+satellite (accessed on 11 January 2023)	6435	36	6	215
Madelon	https://github.com/jundongl/scikit-feature (accessed on 11 January 2023)	2600	500	2	260
Splice	https://archive.ics.uci.edu/dataset/69/molecular+biology+splice+junction+gene+sequences (accessed on 11 January 2023)	3175	60	3	265
Waveform	https://archive.ics.uci.edu/dataset/108/waveform+database+generator+version+2 (accessed on 11 January 2023)	5000	40	3	333
Krvskp	https://archive.ics.uci.edu/mL/datasets/Chess+(King-Rook+vs.+King-Pawn) (accessed on 11 January 2023)	3196	36	2	533
Musk	https://archive.ics.uci.edu/dataset/75/musk+version+2 (accessed on 11 January 2023)	6598	166	2	660

**Table 3 entropy-25-01003-t003:** Accuracy (%) of selected features using the k-nearest neighbors (KNN) algorithm.

No.	Dataset	CONDRED	MIM	mRMR	DISR	CIFE	WBFS
1	Lung	46.43 ± 16.94 *	77.68 ± 18.08 *	85.71 ± 15.06	86.07 ± 14.72	61.25 ± 16.79 *	**88.93 ± 8.94**
2	Breast	93.32 ± 3.07 *	**95.61 ± 2.77**	95.08 ± 1.61	93.33 ± 3.94 *	92.79 ± 2.67 *	95.43 ± 2.06
3	Colon	83.81 ± 13.65	87.14 ± 13.05	**90.24 ± 11.53 v**	83.81 ± 13.65	87.14 ± 6.85	85.48 ± 12.27
4	Ionosphere	85.46 ± 5.14	**86.32 ± 4.24**	85.44 ± 4.97	86.32 ± 5.02 **v**	85.17 ± 6.17	84.60 ± 5.11
5	Isolet	20.06 ± 2.74 *	29.74 ± 3.50 *	**65.06 ± 4.29 v**	57.69 ± 3.32 *	50.90 ± 4.00 *	59.94 ± 3.45
6	Krvskp	88.95 ± 1.69 *	96.25 ± 0.94	94.15 ± 1.26 *	93.37 ± 1.04 *	94.4 ± 1.27 *	**96.25 ± 0.94**
7	Landsat	84.51 ± 1.54	83.03 ± 1.12 *	84.79 ± 1.10	84.16 ± 1.64	**85.30 ± 1.16**	84.48 ± 1.76
8	Madelon	76.38 ± 2.43 *	76.85 ± 2.18 *	57.5 ± 3.82 *	81.42 ± 1.78 *	81.65 ± 3.07	**82.92 ± 1.84**
9	Musk	50.85 ± 2.18 *	87.06 ± 1.00 *	83.13 ± 0.95 *	69.25 ± 1.82 *	76.66 ± 1.82 *	**91.66 ± 1.37**
10	Sonar	75.45 ± 9.19	83.17 ± 10.51	78.33 ± 8.59	**83.64 ± 9.12**	80.24 ± 8.99	81.17 ± 10.29
11	Splice	**82.87 ± 1.78**	82.43 ± 2.02	82.43 ± 2.02	82.43 ± 2.02	79.97 ± 2.47 *	82.43 ± 2.02
12	Waveform	80.52 ± 1.44	80.52 ± 1.44	80.52 ± 1.44	80.52 ± 1.44	69.94 ± 2.02 *	**80.52 ± 1.44**
Average		72.38	80.48	81.87	81.83	78.78	**84.48**
W/T/L		6/6/0	5/7/0	3/7/2	5/6/1	7/5/0	

The ‘*’ symbols and ‘v’ identify statistically significant (at the 0.1 level) wins or losses over the proposed WBFS method.

**Table 4 entropy-25-01003-t004:** Accuracy (%) of selected features using the naïve Bayes classifier (NBC) algorithm.

No.	Dataset	CONDRED	MIM	mRMR	DISR	CIFE	WBFS
1	Lung	51.96 ± 24.43	58.75 ± 24.69	61.61 ± 23.35	61.61 ± 23.35	60.18 ± 22.59	60.18 ± 22.11
2	Breast	89.99 ± 3.60 *	92.79 ± 3.14	93.32 ± 2.84	62.74 ± 6.29 *	92.79 ± 2.26	**93.32 ± 2.96**
3	Colon	80.48 ± 10.75 *	88.57 ± 13.65	88.57 ± 13.65	88.57 ± 13.65	69.05 ± 18.58 *	**88.81 ± 11.06**
4	Ionosphere	72.66 ± 9.70 *	64.10 ± 6.04 *	64.10 ± 6.04 *	64.10 ± 6.04 *	80.35 ± 10.45 *	**84.62 ± 7.75**
5	Isolet	22.24 ± 3.96 *	24.17 ± 2.99 *	26.92 ± 4.46 *	16.79 ± 2.80 *	**55.83 ± 3.29 v**	43.08 ± 4.35
6	Krvskp	52.22 ± 1.88	52.22 ± 1.88	52.22 ± 1.88	52.22 ± 1.88	52.22 ± 1.88	**52.22 ± 1.88**
7	Landsat	75.46 ± 1.71 *	72.82 ± 1.76 *	76.53 ± 1.85	76.64 ± 1.52	**77.62 ± 2.16 v**	76.38 ± 1.94
8	Madelon	59.58 ± 3.10	**59.62 ± 2.36 v**	58.96 ± 2.31	59.19 ± 2.52	59.15 ± 3.65	59.12 ± 2.15
9	Musk	73.78 ± 1.49 *	77.39 ± 1.79 *	61.56 ± 1.60 *	82.69 ± 0.96 **v**	**88.88 ± 0.93 v**	78.87 ± 1.35
10	Sonar	64.79 ± 13.19 *	70.12 ± 15.74	66.74 ± 16.06	68.67 ± 15.47 *	71.57 ± 9.61	**73.48 ± 16.34**
11	Splice	**88.69 ± 8.38**	88.62 ± 8.37	88.62 ± 8.37	88.62 ± 8.37	87.74 ± 7.96 *	88.62 ± 8.37
12	Waveform	79.08 ± 1.26	79.08 ± 1.26	79.08 ± 1.26	79.08 ± 1.26	77.34 ± 0.91 *	**79.08 ± 1.26**
Average		67.58	69.02	68.19	66.74	72.73	**73.15**
W/T/L		7/5/0	4/7/1	3/9/0	4/7/1	4/5/3	

The ‘*’ symbols and ‘v’ identify statistically significant (at the 0.1 level) wins or losses over the proposed WBFS method.

**Table 5 entropy-25-01003-t005:** Accuracy (%) of selected features using the Library for Support Vector Machines (LibSVM) algorithm.

No.	Dataset	CONDRED	MIM	mRMR	DISR	CIFE	WBFS
1	Lung	43.75 ± 24.12	46.25 ± 24.69 *	41.25 ± 27.36	**46.25 ± 24.69 v**	28.93 ± 23.28 *	42.5 ± 24.09
2	Breast	93.32 ± 2.60 *	94.38 ± 2.72	95.25 ± 1.90	92.97 ± 2.35 *	92.79 ± 3.25 *	**95.42 ± 2.40**
3	Colon	62.86 ± 22.14	**69.29 ± 19.97 v**	64.52 ± 23.07	64.52 ± 23.07	64.52 ± 23.07	64.52 ± 23.07
4	Ionosphere	83.47 ± 6.99 *	88.60 ± 4.48	**89.16 ± 4.24 v**	88.90 ± 4.72 *	87.46 ± 4.71	87.18 ± 5.25
5	Isolet	28.33 ± 2.36 *	33.78 ± 4.34 *	**68.21 ± 2.92 v**	61.15 ± 1.87 *	61.79 ± 2.59 *	64.68 ± 3.46
6	Krvskp	75.28 ± 2.55 *	77.63 ± 2.10	72.37 ± 2.39 *	72.03 ± 2.38 *	72.59 ± 2.03 *	**77.63 ± 2.10**
7	Landsat	82.77 ± 1.40	80.57 ± 1.56 *	82.78 ± 1.40	82.81 ± 1.62	**83.37 ± 1.35 v**	82.69 ± 1.50
8	Madelon	65.85 ± 1.41 *	65.62 ± 1.28 *	60.35 ± 1.57 *	67.77 ± 1.94 *	67.81 ± 2.56 *	**69.50 ± 2.39**
9	Musk	84.59 ± 1.27 *	84.60 ± 1.23 *	84.59 ± 1.27 *	84.59 ± 1.27 *	90.21 ± 0.93 *	**91.12 ± 1.10**
10	Sonar	64.83 ± 8.47 *	74.00 ± 12.71 *	76.4 ± 13.56	74.95 ± 12.35	75.40 ± 10.31	**76.88 ± 11.64**
11	Splice	**89.39 ± 2.23**	89.17 ± 2.07	89.17 ± 2.07	89.17 ± 2.07	87.62 ± 1.78 *	89.17 ± 2.07
12	Waveform	83.62 ± 1.97	83.62 ± 1.97	83.62 ± 1.97	83.62 ± 1.97	77.32 ± 1.10 *	**83.62 ± 1.97**
Average		72.38	80.48	81.87	81.83	78.78	**83.01**
W/T/L		7/5/0	6/5/1	3/7/2	6/5/1	8/3/1	

The ‘*’ symbols and ‘v’ identify statistically significant (at the 0.1 level) wins or losses over the proposed WBFS method.

**Table 6 entropy-25-01003-t006:** The Receiver Operating Characteristic (ROC) analysis of 10 selected protein biomarkers and a combination test of 8 biomarkers.

No.	Protein Biomarker	AUC	Sensitivity	Specificity	*p*-Value
1	**BRD4**	0.749	59.38	82.60	<0.0001
2	**CD26**	0.824	75.70	75.40	<0.0001
3	**DUSP4**	0.733	67.40	67.40	<0.0001
4	**GAPDH**	0.775	73.50	67.70	<0.0001
5	GSK3ALPHABETA	0.765	66.50	75.40	<0.0001
6	IGFBP2	0.722	65.20	71.80	<0.0001
7	**INPP4B**	0.767	74.20	66.90	<0.0001
8	**MIG6**	0.833	80.00	72.70	<0.0001
9	**NDRG1_pT346**	0.729	62.20	76.50	<0.0001
10	**TFRC**	0.876	81.20	80.90	<0.0001
11	The combination of eight biomarkers	0.960	90.15	92.54	<0.0001

The detailed information of the eight biomarkers in combination test are shown in Appendix A.

## Data Availability

The data presented in this study are available at https://github.com/jihanwang/WBFS (accessed on 11 January 2023).

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
