# Peer review of "The Weight-Based Feature Selection (WBFS) Algorithm Classifies Lung Cancer Subtypes Using Proteomic Data"

_entropy, 2023, doi:10.3390/e25071003_

Round 1

Reviewer 1 Report

This manuscript aims to develop a feature selection method BWFS. Then the authors used this method to compare with other feature selection or classification methods. Finally, by using BWFS, the authors identified ten markers as potential biomarkers for LUAD and LUSC sample types. 

First, it is unclear why the authors want to use the protein-level expression to test the BWFS method, but not the RNA expression data, which are more commonly used and since there are multiple studies on lung cancer markers, it might be easier for confirmation. 

Second, the BWFS method was described with sufficient details but the current content structure and organization are confusing. If the authors want to emphasize the ten biomarkers for lung cancer subtypes, then they should focus on the validation of these two markers and try to associate them with established studies. 

Third, when describing the LUAD and LUSC data tSNE and PCA results, is it possible that the difference is just driven by the batch effect between datasets? Have the authors conducted any batch effect correction or normalization? More details are required as to how datasets are imported. 

Next, what is the parameter ratio in Table 2? How are the ratios derived or are they some intrinsic parameters per dataset? And from the comparison between BWFS and other methods, MIM seems to be superior to BWFS in several datasets but far inferior in others. Have the authors considered if any structures of the testing dataset could affect algorithm performance? And when comparing the algorithm performances, would it be another significant factor to also compare the computational complexities? For example, would it be significant to sacrifice <1% of accuracy for much more computational resources? 

Another minor comment, since this method is developed/described to identify biomarkers, it is curious whether or not the objection functions bear any biological meanings. 

Author Response

The attachment is our reply, thank you

Reviewer 2 Report

Thank you for submitting your manuscript entitled "Using BWFS method for classifying lung cancer subtypes". After a thorough review of your submission, I have the following comments.

Your manuscript used and compared several feature selection methods to classify the same broad lung cancer into two at the cell levels i.e., LUSC vs LUAD. The authors used the TCGA/TCPA data based on transcriptomics data and used BWFS and Bayesian networks to classify the tumor at different cell morphology levels. Feature selection is a critical method and application for building any type of machine learning. These methods help identify potential features that could be used for biomarkers for early diagnosis, prognosis, and therapeutics.

The overall manuscript design, writing, arrangement, and contents look appropriate for publication. However, I would like to suggest that the authors revise the manuscript extensively before publication. In my experience, several publications have come out using similar methods, and the authors need to ensure that their work is novel and significant in this area.

Comment 1 : Need to change the titles. The titles are not representing the entire study. From current titles, I'm confused that you used BWFS methods for the classification of lung cancer based on the transcriptomics data or the imaging data or alleles or SNPs based. If you mention like, what kind of data are using to classify through this method would be great. 

Comment 2 : Can you abbreviate the "BWFS" in the title? 

Comment 3 : Line no. 11-12, change "In this paper" to "In this study"

comment 4 : Line no. 12, "We have applied"

Comment 5 : Change the words "novel" to "unique" , and established a new method. 

comment 6 : change "obtain" to "identify" in line no 13. 

comment 7 : change "causalities" to "survival analysis"

comment 8 : change "critical biomarkers" to "potential biomarkers" in line no. 16. 

comment 9 : Please don't use "novel" to "new or something else words" in line no.18.  I would recommend use some different words

comment 10: Add "TCGA and Lung Cancer" in the keywords and remove "interaction information"

comment 11 : Remove the "two of the" from line no. 26.

Comment 12 : Line no 41 and 41. Add the following reference. 

comment 13 : change Nobel to unique in line no 75. 

Comment 14 : Use the PCA analysis rather than tSNE and put the PCA figure into the supplementary and give the proper reason why t-SNE is used here. 

Comment 15 : check all the theorem thoroughly and all mathematical equations

Comment 16 : How many protein biomarkers were identified using your algorithms? and how many biomarkers have been used for the PCA/t-SNE? 

Comment 17: Authors have used paid software like MATLAB. Matlab is not available to every lab, the producibility of your study would be less because of the paid software. I would recommend using the open access tools such as Python or R to reproduce your study and submit your code on GitHub. 

Comment 18. The authors has also not used the most common feature selection methods like forward and backward selections, Random Forest, etc,

Comment 19 : I didn't find any significant difference between your algorithm's classification rate vs other established or known algorithms. I would recommend please the hypothesis testing of the accuracy prediction so that the reviewer and reader can understand the significant of the classification of BWFs vs others. 

Comment 20 : Add the DOI 10.7717/peerj.9656 and doi: 10.3389/fonc.2022.910494 in your discussion part. 

Comment 21 : The manuscript need to revise extensively and resubmit with all comments.

Author Response

The attachment is our reply, thank you

Round 2

Reviewer 1 Report

The authors have adequately addressed all the previous comments and provided a detailed description of the method and analysis. The overall clarity of the manuscript has been greatly improved. 

Author Response

Thank you for taking the time to review the revised version of our manuscript. We sincerely appreciate your positive feedback and acknowledgment of the improvements made.

We are grateful for your thorough review and valuable input, which have undoubtedly contributed to the overall improvement of our manuscript. Your feedback has been instrumental in refining our work, and we are grateful for your guidance throughout this process.

Reviewer 2 Report

Thank you for addressing all comments and concerns about the manuscript. Now, it's looking much better and easy to understand for the general public or reader. However, I found some of the work need to perform before publication. Here are some potential comments. 

1) Can you transcribe your Matlab script/command/performed work in the open access library such as R, python. 

2) Authors haven't addressed the general feature selection model comparison such as Random Forest. 

3) Thoroughly check the spelling, grammar mistakes, and figures and tables sequence in the manuscript. 

4) Check plagiarism if found then reduce it.  

Need to improve a little bit. 

Author Response

We thank the reviewer for the patient and meticulous guidance and suggestions.

Round 3

Reviewer 2 Report

The authors have addressed all the asked comments and now the manuscript can be accepted for publication. Good luck and all the best

Author Response

We thank the reviewer's all valuable comments.
